# Twin Auxiliary Classifiers GAN

Mingming Gong [*1,3], Yanwu Xu [*1], Chunyuan Li[2], Kun Zhang[3], and Kayhan Batmanghelich[1]

[1]Department of Biomedical Informatics, University of Pittsburgh, {mig73,yanwuxu,kayhan}@pitt.edu
[2]Microsoft Research, Redmond, cl319@duke.edu
[3]Department of Philosophy, Carnegie Mellon University, kunz1@cmu.edu

## Abstract

Conditional generative models enjoy remarkable progress over the past few years. One of the popular conditional models is Auxiliary Classifier GAN (AC-GAN), which generates highly discriminative images by extending the loss function of GAN with an auxiliary classifier. However, the diversity of the generated samples by AC-GAN tends to decrease as the number of classes increases, hence limiting its power on large-scale data. In this paper, we identify the source of the low diversity issue theoretically and propose a practical solution to solve the problem. We show that the auxiliary classifier in AC-GAN imposes perfect separability, which is disadvantageous when the supports of the class distributions have significant overlap. To address the issue, we propose Twin Auxiliary Classifiers Generative Adversarial Net (TAC-GAN) that further benefits from a new player that interacts with other players (the generator and the discriminator) in GAN. Theoretically, we demonstrate that TAC-GAN can effectively minimize the divergence between the generated and real-data distributions. Extensive experimental results show that our TAC-GAN can successfully replicate the true data distributions on simulated data, and significantly improves the diversity of class-conditional image generation on real datasets.

## 1  Introduction

Generative Adversarial Networks (GANs) [1] are a framework to learn the data generating distribution implicitly. GANs mimic sampling from a target distribution by training a generator that maps samples drawn from a canonical distribution to the data space. A distinctive feature of GANs is that the discriminator that evaluates the separability of the real and generated data distributions [1–4]. If the discriminator can hardly distinguish between real and generated data, the generator is likely to provide a good approximation to the true data distribution. To generate high-fidelity images, much recent research has focused on designing more advanced network architectures [5, 6], developing more stable objective functions [7–9, 3], enforcing appropriate constraints on the discriminator [10–12], or improving training techniques [7, 13].

Conditional GANs (cGANs) [14] are a variant of GANs that take advantage of extra information (condition) and have been widely used for generation of class-conditioned images [15–18] and text [19, 20]. A major difference between cGANs and GANs is that the cGANs feed the condition to both the generator and the discriminator to lean the joint distributions of images and the condition random variables. Most methods feed the conditional information by concatenating it (or its embedding) with the input or the feature vector at specific layers [14, 21, 15, 22, 23, 20]. Recently, Projection-cGAN [24] improves the quality of the generated images using a specific discriminator that takes the inner product between the embedding of the conditioning variable and the feature vector of the input

---

[*]Equal Contribution
The code is available at `https://github.com/batmanlab/twin_ac`.

image, obtaining state-of-the-art class-conditional image generation on large-scale datasets such as ImageNet1000 [25].

Among cGANs, the Auxiliary Classifier GAN (AC-GAN) has received much attention due to its simplicity and extensibility to various applications [17]. AC-GAN incorporates the conditional information (label) by training the GAN discriminator with an additional classification loss. AC-GAN is able to generate high-quality images and has been extended to various learning problems, such as text-to-image generation [26]. However, it is reported in the literature [17, 24] that as the number of labels increases, AC-GAN tends to generate near-identical images for most classes. Miyato *et al.* [24] observed this phenomenon on ImageNet1000 and conjectured that the auxiliary classifier might encourage the generator to produce less diverse images so that they can be easily discernable.

Despite these insightful findings, the exact source of the low-diversity problem is unclear, let alone its remedy. In this paper, we aim to provide an understanding of this phenomenon and accordingly develop a new method that is able to generate diverse and realistic images. First, we show that due to a missing term in the objective of AC-GAN, it does not faithfully minimize the divergence between real and generated conditional distribution. We show that missing that term can result in a degenerate solution, which explains the lack of diversity in the generated data. Based on our understanding, we introduce a new player in the min-max game of AC-GAN that enables us to estimate the missing term in the objective. The resulting method properly estimates the divergence between real and generated conditional distributions and significantly increases sample diversity within each class. We call our method Twin Auxiliary Classifiers GAN (TAC-GAN) since the new player is also a classifier. Compared to AC-GAN, our TAC-GAN successfully replicates the real data distributions on simulated data and significantly improves the quality and diversity of the class-conditional image generation on CIFAR100 [27], VGGFace2 [28], and ImageNet1000 [25] datasets. In particular, to our best knowledge, our TAC-GAN is the first cGAN method that can generate good quality images on the VGGFace dataset, demonstrating the advantage of TAC-GAN on fine-grained datasets.

## 2  Method

In this section, we review the Generative Adversarial Network (GAN) [1] and its conditional variant (cGAN) [14]. We review one of the most popular variants of cGAN, which is called Auxiliary Classifier GAN (AC-GAN) [17]. We first provide an understanding of the observation of low-diversity samples generated by AC-GAN from a distribution matching perspective. Second, based on our new understanding of the problem, we propose a new method that enables learning of real distributions and increasing sample diversity.

### 2.1  Background

Given a training set $\{\mathbf{x}_i\}_{i=1}^n \subseteq \mathcal{X}$ drawn from an unknown distribution $P_X$, GAN estimates $P_X$ by specifying a distribution $Q_X$ implicitly. Instead of an explicit parametrization, it trains a generator function $G(Z)$ that maps samples from a canonical distribution, i.e., $Z \sim P_Z$, to the training data. The generator is obtained by finding an equilibrium of the following mini-max game that effectively minimizes the Jensen-Shannon Divergence (JSD) between $Q_X$ and $P_X$:

$$\min_G \max_D \mathop{\mathbb{E}}_{X \sim P_X}[\log D(X)] + \mathop{\mathbb{E}}_{Z \sim P_Z}[\log(1 - D(G(Z)))], \tag{1}$$

where $D$ is a discriminator. Notice that the $Q_X$ is not directly modeled.

Given a pair of observation ($\mathbf{x}$) and a condition ($y$), $\{\mathbf{x}_i, y\}_{i=1}^n \subseteq \mathcal{X} \times \mathcal{Y}$ drawn from the joint distribution $(\mathbf{x}_i, y) \sim P_{XY}$, the goal of cGAN is to estimate a conditional distribution $P_{X|Y}$. Let $Q_{X|Y}$ denote the conditional distribution specified by a generator $G(Y, Z)$ and $Q_{XY} := Q_{X|Y} P_Y$. A generic cGAN trains $G$ to implicitly minimize the JSD divergence between the joint distributions $Q_{XY}$ and $P_{XY}$:

$$\min_G \max_D \mathop{\mathbb{E}}_{(X,Y) \sim P_{XY}}[\log D(X, Y)] + \mathop{\mathbb{E}}_{Z \sim P_Z, Y \sim P_Y}[\log(1 - D(G(Z, Y), Y))]. \tag{2}$$

In general, $Y$ can be a continuous or discrete variable. In this paper, we focus on case that $Y$ is the (discrete) class label, *i.e.*, $\mathcal{Y} = \{1, \ldots, K\}$.

## 2.2 Insight on Auxiliary Classifier GAN (AC-GAN)

AC-GAN introduces a new player $C$ which is a classifier that interacts with the $D$ and $G$ players. We use $Q^c_{Y|X}$ to denote the conditional distribution induced by $C$. The AC-GAN optimization combines the original GAN loss with cross-entropy classification loss:

$$\min_{G,C} \max_{D} \ \mathcal{L}_{\text{AC}}(G,D,C) = \underbrace{\mathbb{E}_{X \sim P_X}[\log D(X)] + \mathbb{E}_{Z \sim P_Z, Y \sim P_Y}[\log(1 - D(G(Z,Y)))]}_{\text{\textcircled{a}}}$$

$$- \lambda_c \underbrace{\mathbb{E}_{(X,Y) \sim P_{XY}}[\log C(X,Y)]}_{\text{\textcircled{b}}} - \lambda_c \underbrace{\mathbb{E}_{Z \sim P_Z, Y \sim P_Y}[\log(C(G(Z,Y),Y))]}_{\text{\textcircled{c}}}, \quad (3)$$

where $\lambda_c$ is a hyperparameter balancing GAN and auxiliary classification losses.

Here we decompose the objective of AC-GAN into three terms. Clearly, the first term ⓐ corresponds to the Jensen-Shannon divergence (JSD) between $Q_X$ and $P_X$. The second term ⓑ is the cross-entropy loss on real data. It is straightforward to show that the second term minimizes Kullback-Leibler (KL) divergence between the real data distribution $P_{Y|X}$ and the distribution $Q^c_{Y|X}$ specified by $C$. To show that, we can add the negative conditional entropy, $-H_P(Y|X) = \mathbb{E}_{(X,Y) \sim P_{XY}}[\log P(Y|X)]$, to ⓑ, we have

$$-H_P(Y|X) + \text{\textcircled{b}} = \mathbb{E}_{(X,Y) \sim P_{XY}}[\log P(Y|X)] - \mathbb{E}_{(X,Y) \sim P_{XY}}[\log C(X,Y)]$$

$$= \mathbb{E}_{(X,Y) \sim P_{XY}}[\log P(Y|X)] - \mathbb{E}_{(X,Y) \sim P_{XY}}[\log Q^c(Y|X)]$$

$$= \mathbb{E}_{(X,Y) \sim P_{XY}}\left[\log \frac{P(Y|X)}{Q^c(Y|X)}\right] = \text{KL}(P_{Y|X}||Q^c_{Y|X}). \quad (4)$$

Since the negative conditional entropy $-H_P(Y|X)$ is a constant term, minimizing ⓑ w.r.t. the network parameters in $C$ effectively minimizes the KL divergence between $P_{Y|X}$ and $Q^c_{Y|X}$.

The third term ⓒ is the cross-entropy loss on the generated data. Similarly, *if* one adds the negative entropy $-H_Q(Y|X) = \mathbb{E}_{(X,Y) \sim Q_{XY}}[\log Q(Y|X)]$ to ⓒ and obtain the following result:

$$-H_Q(Y|X) + \text{\textcircled{c}} = \mathbb{E}_{(X,Y) \sim Q_{XY}}[\log Q(Y|X)] - \mathbb{E}_{(X,Y) \sim Q_{XY}}[\log Q^c(Y|X)] = \text{KL}(Q_{Y|X}||Q^c_{Y|X}).$$

When updating $C$, $-H_Q(Y|X)$ can be considered a constant term, thus minimizing ⓒ w.r.t. $C$ effectively minimizes the KL divergence between $Q_{Y|X}$ and $Q^c_{Y|X}$. However, when updating $G$, $-H_Q(Y|X)$ cannot be considered as a constant term, because $Q_{Y|X}$ is the conditional distribution specified by the generator $G$. AC-GAN ignores $-H_Q(Y|X)$ and only minimizes ⓒ when updating $G$ in the optimization procedure, which fails to minimize the KL divergence between $Q_{Y|X}$ and $Q^c_{Y|X}$. We hypothesize that the likely reason behind low diversity samples generated by AC-GAN is that it fails to account for the missing term while updating $G$. In fact, the following theorem shows that AC-GAN can converge to a degenerate distribution:

**Theorem 1.** *Suppose $P_X = Q_X$. Given an auxiliary classifier $C$ which specifies a conditional distribution $Q^c_{Y|X}$, the optimal $G^*$ that minimizes ⓒ induces the following degenerate conditional distribution $Q^*_{Y|X}$,*

$$Q^*(Y = k|X = x) = \begin{cases} 1, & \text{if } k{=}\arg\max_i Q^c(Y = i|X = x), \\ 0, & \text{otherwise.} \end{cases} \quad (5)$$

Proof is given in Section S1 of the Supplementary Material (SM). Theorem 1 shows that, even when the marginal distributions are perfectly matched by GAN loss ⓐ, AC-GAN is not able to model the probability when class distributions have support overlaps. It tends to generate data in which $Y$ is deterministically related to $X$. This means that the generated images for each class are confined by the regions induced by the decision boundaries of the auxiliary classifier $C$, which fails to replicate conditional distribution $Q^c_{Y|X}$, implied by $C$, and reduces the distributional support of each class. The theoretical result is consistent with the empirical results in [24] that AC-GAN generates discriminable images with low intra-class diversity. It is thus essential to incorporate the missing term, $-H_Q(Y|X)$, in the objective to penalize this behavior and minimize the KL divergence between $Q_{Y|X}$ and $Q^c_{Y|X}$.

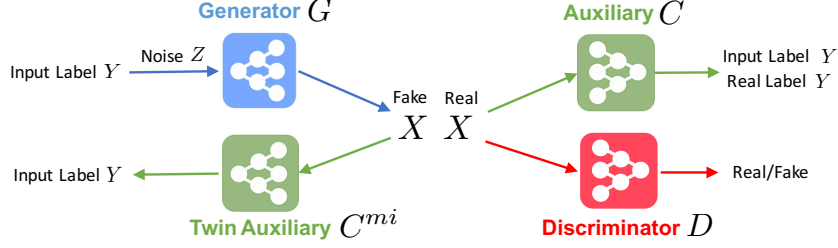

Figure 1: Illustration of the proposed TAC-GAN. The generator $G$ synthesizes fake samples $X$ conditioned input label $Y$. The discriminator $D$ distinguishes between real/fake samples. The auxiliary classifier $C$ is trained to classify labels on both real and fake pairs, while the proposed twin auxiliary classifier $C^{mi}$ is trained on fake pairs only.

## 2.3 Twin Auxiliary Classifiers GAN (TAC-GAN)

Our analysis in the previous section motivates adding the missing term, $-H_Q(Y|X)$, back to the objective function. While minimizing ⓒ forces $G$ to concentrate the conditional density mass on the training data, $-H_Q(Y|X)$ works in the opposite direction by increasing the entropy. However, estimating $-H_Q(Y|X)$ is a challenging task since we do not have access to $Q_{Y|X}$. Various methods have been proposed to estimate the (conditional) entropy, such as [29–32]; however, these estimators cannot be easily used as an objective function to learn $G$ via backpropagation. Below we propose to estimate the conditional entropy by adding a new player in the mini-max game.

The general idea is to introduce an additional auxiliary classifier, $C^{mi}$, that aims to identify the labels of the samples drawn from $Q_{X|Y}$; the low-diversity case makes this task easy for $C^{mi}$. Similar to GAN, the generator tries to compete with the $C^{mi}$. The overall idea of TAC-GAN is illustrated in Figure 1. In the following, we demonstrate its connection with minimizing $-H_Q(Y|X)$.

**Proposition:** Let us assume, without loss of generality, that all classes are equally likely [1] (*i.e.*, $P(Y = k) = \frac{1}{K}$). Minimizing $-H_Q(Y|X)$ is equivalent to minimizing (1) the mutual information between $Y$ and $X$ and (2) the JSD between the conditional distributions $\{Q_{X|Y=1}, \ldots, Q_{X|Y=K}\}$.

*Proof.*

$$
\begin{aligned}
I_Q(Y, X) &= H(Y) - H_Q(Y|X) = H_Q(X) - H_Q(X|Y) \\
&= -\frac{1}{K}\sum_{k=1}^{K} \mathop{\mathbb{E}}_{X \sim Q_{X|Y=k}} \log Q(X) + \frac{1}{K}\sum_{k=1}^{K} \mathop{\mathbb{E}}_{X \sim Q_{X|Y=k}} \log Q(X|Y=k) \\
&= \frac{1}{K}\sum_{k=1}^{K} \mathrm{KL}(Q_{X|Y=k} || Q_X) = \mathrm{JSD}(Q_{X|Y=1}, \ldots, Q_{X|Y=K}). \quad (6)
\end{aligned}
$$

(1) follows from the fact that entropy of $Y$ is constant with respect to $Q$, (2) is shown above. $\qquad\square$

Based on the connection between $-H_Q(Y|X)$ and JSD, we extend the two-player minimax approach in GAN [1] to minimize the JSD between multiple distributions. More specifically, we use another auxiliary classifier $C^{mi}$ whose last layer is a softmax function that predicts the probability of $X$ belong to a class $Y = k$. We define the following minimax game:

$$
\min_{G} \max_{C^{mi}} V(G, C^{mi}) = \mathop{\mathbb{E}}_{Z \sim P_Z, Y \sim P_Y} [\log(C^{mi}(G(Z, Y), Y))]. \quad (7)
$$

The following theorem shows that the minimax game can effectively minimize the JSD between $\{Q_{X|Y=1}, \ldots, Q_{X|Y=K}\}$.

**Theorem 2.** *Let* $U(G) = \max_{C^{mi}} V(G, C^{mi})$. *The global mininum of the minimax game is achieved if and only if* $Q_{X|Y=1} = Q_{X|Y=2} = \cdots = Q_{X|Y=K}$. *At the optimal point,* $U(G)$ *achieves the value* $-K \log K$.

A complete proof of Theorem 2 is given in Section S2 of the SM. It is worth noting that the global optimum of $U(G)$ cannot be achieved in our model because of other terms in our TAC-GAN objective function, which is obtained by combing (7) and the original AC-GAN objective (3):

$$\min_{G,C} \max_{D,C^{mi}} \mathcal{L}_{\text{TAC}}(G, D, C, C^{mi}) = \mathcal{L}_{\text{AC}}(G, D, C) + \lambda_c V(G, C^{mi}). \qquad (8)$$

The following theorem provides approximation guarantees for the joint distribution $P_{XY}$, justifying the validity of our proposed approach.

**Theorem 3.** *Let $P_{YX}$ and $Q_{YX}$ denote the data distribution and the distribution specified by the generator $G$, respectively. Let $Q^c_{Y|X}$ denote the conditional distribution of $Y$ given $X$ specified by the auxiliary classifier $C$. We have*

$$JSD(P_{XY}, Q_{XY}) \leq 2c_1\sqrt{2JSD(P_X, Q_X)} + c_2\sqrt{2KL(P_{Y|X}||Q^c_{Y|X})} + c_2\sqrt{2KL(Q_{Y|X}||Q^c_{Y|X})},$$

where $c_1$ and $c_2$ are upper bounds of $\frac{1}{2}\int|P_{Y|X}(y|x)|\mu(x,y)$ and $\frac{1}{2}\int|Q_X(x)|\mu(x)$ ($\mu$ is a $\sigma$-finite measure), respectively. A proof of Theorem 3 is provided in Section S3 of the SM.

## 3 Related Works

**TAC-GAN learns an unbiased distribution.** Shu *et al.* [33] first show that AC-GAN tends to down-sample the data points near the decision boundary, causing a biased estimation of the true distribution. From a Lagrangian perspective, they consider AC-GAN as minimizing $JSD(P_X, Q_X)$ with constraints enforced by classification losses. If $\lambda_c$ is very large such that $JSD(P_X, Q_X)$ become less effective, the generator will push the generated images away from the boundary. However, on real datasets, we can also observe low diversity when $\lambda_c$ is small, which cannot be explained by the analysis in [33]. We take a different perspective by constraining $JSD(P_X, Q_X)$ to be small and investigate the properties of the conditional $Q_{Y|X}$. Our analysis suggests that even when $JSD(P_X, Q_X) = 0$, the AC-GAN cross-entropy loss can still result in biased estimate of $Q_{Y|X}$, reducing the support of each class in the generated distribution, compared to the true distribution. Furthermore, we propose a solution that can remedy the low diversity problem based on our understandings.

**Connecting TAC-GAN with Projection cGAN.** AC-GAN was once the state-of-the-art method before the advent of Projection cGAN [24]. Projection cGAN, AC-GAN, and our TAC-GAN share the similar spirits in that image generation performance can be improved when the joint distribution matching problem is decomposed into two easier sub-problems: marginal matching and conditional matching [32]. Projection cGAN decomposes the density ratio, while AC-GAN and TAC-GAN directly decompose the distribution. Both Projection cGAN and TAC-GAN are theoretically sound when using the cross-entropy loss. However, in practice, hinge loss is often preferred for real data. In this case, Projection cGAN loses the theoretical guarantee, while TAC-GAN is less affected, because only the GAN loss is replaced by the hinge loss.

## 4 Experiments

We first compare the distribution matching ability of AC-GAN, Projection cGAN, and our TAC-GAN on Mixture of Gaussian (MoG) and MNIST [34] synthetic data. We evaluate the image generation performance of TAC-GAN on three image datatest including CIFAR100 [27], ImageNet1000 [25] and VGGFace2 [28]. In our implementation, the twin auxiliary classifiers share the same convolutional layers, which means TAC-GAN only adds a negligible computation cost to AC-GAN. The detailed experiment setups are shown in the SM. We implemented TAC-GAN in Pytorch. To illustrate the algorithm, we submit the implementation on the synthetic datasets in SM. The source code to reproduce the full experimental results will be made public on GitHub.

### 4.1 MoG Synthetic Data

We start with a simulated dataset to verify that TAC-GAN can accurately match the target distribution. We draw samples from a one-dimensional MoG distribution with three Gaussian components, labeled

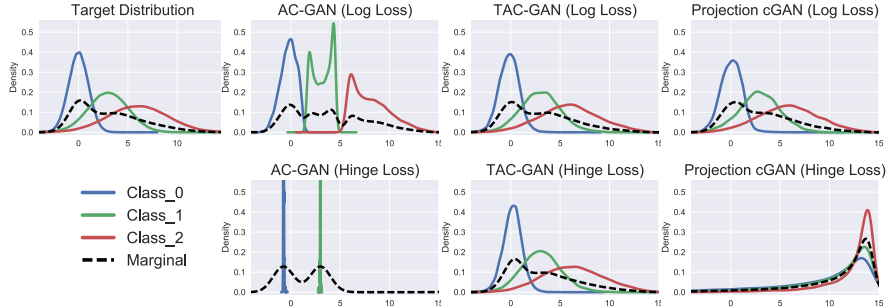

Figure 2: Comparison of sample quality on a synthetic MoG dataset.

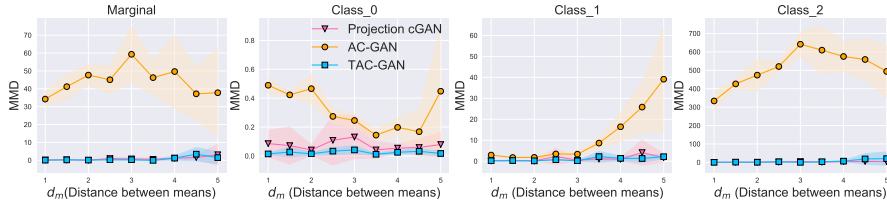

Figure 3: The MMD evaluation. The x-axis means the distance between the means of adjacent Gaussian components ($d_m$). Lower score is better.

as Class_0, Class_1, and Class_2, respectively. The standard deviations of the three components are fixed to $\sigma_0 = 1, \sigma_1 = 2$, and $\sigma_2 = 3$. The differences between the means are set to $\mu_1 - \mu_0 = \mu_2 - \mu_1 = d_m$, in which $d_m$ ranges from 1 to 5. These values are chosen such that the supports of the three distributions have different overlap sizes. We detail the experimental setup in Section S4 of the SM.

Figure 2 shows the ground truth density functions when $\mu_0 = 0, \mu_1 = 3, \mu_2 = 6$ and the estimated ones by AC-GAN, TAC-GAN, and Projection cGAN. The estimated density function is obtained by applying kernel density estimation [35] on the generated data. When using cross-entropy loss, AC-GAN learns a biased distribution where all the classes are perfectly separated by the classification decision function, verifying our Theorem 1. Both our TAC-GAN and Projection cGAN can accurately learn the original distribution. Using Hinge loss, our model can still learn the distribution well, while neither AC-GAN nor Projection cGAN can replicate the real distribution (see Supplementary S4 for more experiments). We also conduct simulation on a 2D dataset and the details are given in Supplementary S5. The results show that our TAC-GAN is able to learn the true data distribution.

Figure 3 reports the Maximum Mean Discrepancy (MMD) [36] distance between the real data and generated data for different $d_m$ values. Here all the GAN models are trained using cross-entropy loss (log loss). The TAC-GAN produces near-zero MMD values for all $d_m$'s, meaning that the data generated by TAC-GAN is very close to the ground truth data. Projection cGAN performs slightly worse than TAC-GAN and AC-GAN generates data that have a large MMD distance to the true data.

## 4.2 Overlapping MNIST

Following experiments in [33] to show that AC-GAN learns a biased distribution, we use the overlapping MNIST dataset to demonstrate the robustness of our TAC-GAN. We randomly sample from MNIST training set to construct two image groups: Group $A$ contains 5,000 digit '1' and 5,000 digit '0', while Group $B$ contains 5,000 digit '2' and 5,000 digit '0',to simulate overlapping distributions, where digit '0' appears in both groups. Note that the ground truth proportion of digit '0', '1' and '2' in this dataset are 0.5, 0.25 and 0.25, respectively.

Figure 4 (a) shows the generated images under different $\lambda_c$. It shows that AC-GAN tends to down sample '0' as $\lambda_c$ increases, while TAC-GAN can always generate '0's in both groups. To quantitatively measure the distribution of generated images, we pre-train a "perfect" classifier on a MNIST subset only containing digit '0', '1', and '2', and use the classifier to predict the labels of the generated data. Figure 4 (b) reports the label proportions for the generated images. It shows that the label proportion produced by TAC-GAN is very close to the ground truth values regardless of $\lambda_c$, while AC-GAN generates less '0's as $\lambda_c$ increases. More results and detail setting are shown in Section S6 of the SM.

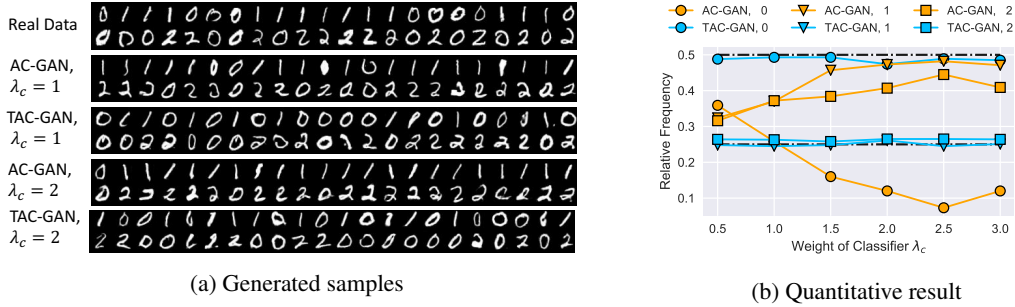

| (a) Generated samples | (b) Quantitative result |

Figure 4: (a) Visualization of the generated MNIST digits with various $\lambda_c$ values. For each section, the top row digits are sampled from group $A$ and the bottom row digits are from group $B$. (b) The label proportion for generated digits of two methods. The ground truth proportion for digit 0,1,2 is [0.5, 0.25, 0.25], visualized as dashed dark lines.

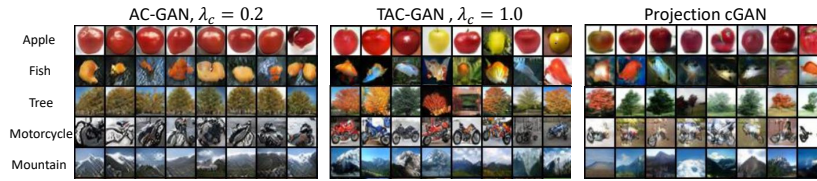

Figure 5: Generated images from five classes of CIFAR100.

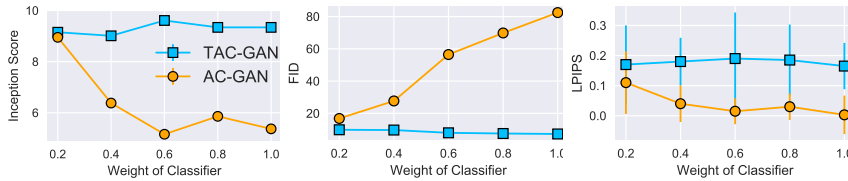

Figure 6: Impact of $\lambda_c$ on the image generation quality on CIFAR100.

## 4.3 CIFAR100

CIFAR100 [27] has 100 classes, each of which contains 500 training images and 100 testing images at the resolution of $32 \times 32$. The current best deep classification model achieves 91.3% accuracy on this dataset [37], which suggests that the class distributions may have certain support overlaps.

Figure 5 shows the generated images for five randomly selected classes. AC-GAN generates images with low intra-class diversity. Both TAC-GAN and Projection cGAN generate visually appealing and diverse images. We provide the generated images for all the classes in Section S6 of the SM.

To quantitatively compare the generated images, we consider the two popular evaluation criteria, including Inception Score (IS) [38] and Fréchet Inception Distance (FID) [39]. We also use the recently proposed Learned Perceptual Image Patch Similarity (LPIPS), which measures the perceptual diversity within each class [40]. The scores are reported in Table 1. TAC-GAN achieves lower FID than Projection cGAN, and outperforms AC-GAN by a large margin, which demonstrates the efficacy of the twin auxiliary classifiers. We report the scores for all the classes in Section S7 of the SM. In Section S7.2 of the SM, we explore the compatibility of our model with the techniques that increase diversity of unsupervised GANs. Specifically, we combine pacGAN [41] with AC-GAN and our TAC-GAN, and the results show that pacGAN can improve both AC-GAN and TAC-GAN, but it cannot fully address the drawbacks of AC-GAN.

**Effects of hyper-parameters $\lambda_c$.** We study the impact of $\lambda_c$ on AC-GAN and TAC-GAN, and report results under different $\lambda_c$ values in Figure 6. It shows that TAC-GAN is robust to $\lambda_c$, while AC-GAN requires a very small $\lambda_c$ to achieve good scores. Even so, AC-GAN generates images with low intra-class diversity, as shown in Figure 5.

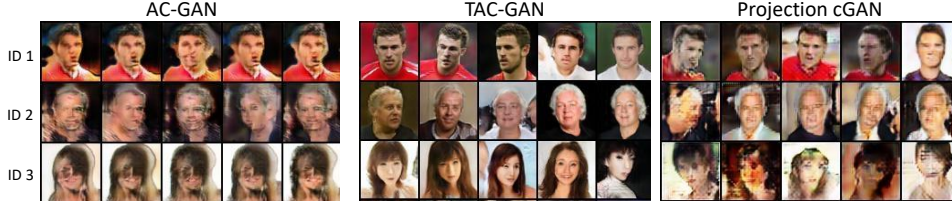

Figure 7: Comparison of generated face samples from three identities in VGGFace2 dataset.

Table 1: The quantitative results of all models on three datasets.

| Methods | AC-GAN ($\lambda_c = 1$) | | TAC-GAN (Ours) ($\lambda_c = 1$) | | Projection cGAN | |
|---|---|---|---|---|---|---|
| Metrics | IS ↑ | FID ↓ | IS ↑ | FID ↓ | IS ↑ | FID ↓ |
| CIFAR100 | $5.37 \pm 0.064$ | 82.45 | $9.34 \pm 0.077$ | **7.22** | **$9.56 \pm 0.133$** | 8.92 |
| ImageNet1000 | $7.26 \pm 0.113$ | 184.41 | $28.86 \pm 0.298$ | 23.75 | **$38.05 \pm 0.790$** | **22.77** |
| VGGFace200 | $27.81 \pm 0.29$ | 95.70 | **$48.94 \pm 0.63$** | **29.12** | $32.50 \pm 0.44$ | 66.23 |
| VGGFace500 | $25.96 \pm 0.32$ | 31.90 | **$77.76 \pm 1.61$** | **12.42** | $35.96 \pm 0.62$ | 43.10 |
| VGGFace1000 | ⌢ | ⌢ | **$108.89 \pm 2.63$** | **13.60** | $71.15 \pm 0.93$ | 24.07 |
| VGGFace2000 | ⌢ | ⌢ | **$109.04 \pm 2.44$** | **13.79** | $79.51 \pm 1.03$ | 22.42 |

## 4.4 ImageNet1000

We further apply TAC-GAN to the large-scale ImageNet dataset [25] containing 1000 classes, each of which has around 1,300 images. We pre-process the data by center-cropping and resizing the images to $128 \times 128$. We detail the experimental setup and attach generated images in Section S8 of the SM.

Table 1 reports the IS and FID metrics of all models. Our TAC-GAN again outperforms AC-GAN by a large margin. In addition, TAC-GAN has lower IS than Projection cGAN. We hypothesize that TAC-GAN has a chance to generate images that do not belong to the given class in the overlap regions, because it aims to model the true conditional distribution.

## 4.5 VGGFace2

VGGFace2 [28] is a large-scale face recognition dataset, with around 362 images for each person. Its main difference to CIFAR100 and ImageNet1000 is that this dataset is more fine-grained with smaller intra-class diversities, making the generative task more difficult. We resize the center-cropped images to $64 \times 64$. To compare different algorithms, we randomly choose 200, 500, 1000 and 2000 identities to construct the VGGFace200, VGGFace500 VGGFACE1000 and VGGFACE2000 datasets, respectively.

Figure 7 shows the generated face images for five randomly selected identities from VGGFACE200. AC-GAN collapses to the class center, generating very similar images for each class. Though Projection cGAN generate diverse images, it has blurry effects. Our TAC-GAN generates diverse and sharp images. To quantitatively compare the methods, we finetune a Inception Net [42] classifier on the face data and then use it to calculate IS and FID score. We report IS and FID scores for all the methods in Table 1. It shows that TAC-GAN produces much better/higher IS better/lower FID score than Projection cGAN, which is consistent with the qualitative observations. These results suggest that TAC-GAN is a promising method for fine-grained datasets. More generated identities are attached in in section S9 of the SM.

## 5 Conclusion

In this paper, we have theoretically analyzed the low intra-class diversity problem of the widely used AC-GAN method from a distribution matching perspective. We showed that the auxiliary classifier in AC-GAN imposes perfect separability, which is disadvantageous when the supports of the class distributions have significant overlaps. Based on the analysis, we further proposed the Twin Auxiliary Classifiers GAN (TAC-GAN) method, which introduces an additional auxiliary classifier to adversarially play with the players in AC-GAN. We demonstrated the efficacy of the proposed

method both theoretically and empirically. TAC-GAN can resolve the issue of AC-GAN to learn an unbiased distribution, and generate high-quality samples on fine-grained image datasets.

**Acknowledgments**

This work was partially supported by NIH Award Number 1R01HL141813-01, NSF 1839332 Tripod+X, and SAP SE. We gratefully acknowledge the support of NVIDIA Corporation with the donation of the Titan X Pascal GPU used for this research. We were also grateful for the computational resources provided by Pittsburgh SuperComputing grant number TG-ASC170024.

## Footnotes

[1]If the dataset is imbalanced, we can apply biased batch sampling to enforce this condition.

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
