[Supplementary Material]

# Supplementary Materials for "Twin Auxiliary Classifiers GAN"

This supplementary material provides the proofs and more experimental details which are omitted in the submitted paper. The equation numbers in this material are consistent with those in the paper.

## S1. Proof of Theorem 1

*Proof.* The optimal $Q^*_{Y|X}$ is obtained by the following optimization problem:

$$\min_{Q_{Y|X}} - \mathop{\mathbb{E}}_{(X,Y)\sim Q_{XY}} [\log Q^c(Y|X)] = - \mathop{\mathbb{E}}_{X\sim Q_X} [\sum_{i=1}^{K} Q(Y=i|X) \log Q^c(Y=i|X)],$$

$$s.t. \sum_{i=1}^{K} Q(Y=i|X=x) = 1 \text{ and } Q(Y=i|X=x) \geq 0. \tag{E1}$$

The optimization problem in (E1) is equivalent to minimizing the objective point-wisely for each $x$, i.e.,

$$\min_{Q_{Y|X=x}} - \sum_{i=1}^{K} Q(Y=i|X=x) \log Q^c(Y=i|X=x),$$

$$s.t. \sum_{i=1}^{K} Q(Y=i|X=x) = 1 \text{ and } Q(Y=i|X=x) \geq 0, \tag{E2}$$

which is a linear programming (LP) problem. The optimal solution must lie in the extreme points of the feasible set, which are those points with posterior probability 1 for one class and 0 for the other classes. By evaluating the objective values of these extreme points, the optimal solution is (5) with objective value $-\log Q^c(Y=k|X=x)$, where $k = \arg\max_i Q^c(Y=i|X=x)$. □

## S2. Proof of Theorem 2

*Proof.* The minimax game (7) can be written as

$$\min_{G} \max_{C^{mi}} V(G, C^{mi}) = \mathop{\mathbb{E}}_{Z\sim P_Z, Y\sim P_Y} [\log(C^{mi}(G(Z,Y),Y))]$$

$$= \mathop{\mathbb{E}}_{X\sim Q_{XY}} [\log(C^{mi}(X,Y))]$$

$$= \frac{1}{K} \sum_{k=1}^{K} \mathop{\mathbb{E}}_{X\sim Q_{X|Y=k}} [\log(C^{mi}(X,Y=k))]$$

$$s.t. \sum_{k=1}^{K} C^{mi}(X,Y=k) = 1, \tag{E3}$$

where the constraint is because $C^{mi}$ is forced to have probability outputs that sum to one. In the following proposition, we will give the optimal $C^{mi}$ for any given $G$, or equivalently $Q_{XY}$.

**Proposition 1.** *Let for a fixed generator $G$, the optimal prediction probabilities $C^{mi}(X=x, Y=k)$ of $C^{mi}$ are*

$$C^{mi*}(x, Y=k) = \frac{Q(x|Y=k)}{\sum_{k'=1}^{K} Q(x|Y=k')}. \tag{E4}$$

*Proof.* For a fixed $G$, (E3) reduces to maximize the value function $V(G, C^{mi})$ w.r.t. $C^{mi}(x, Y = 1), \ldots, C^{mi}(x, Y = K)$:

$$\{C^{mi*}(x, Y = 1), \ldots, C^{mi*}(x, Y = K)\}$$

$$= \arg \max_{C^{mi}(x, Y=1), \ldots, C^{mi}(x, Y=K)} \sum_{k=1}^{K} \int_{x} Q(x|Y = k) \log(C^{mi}(x, Y = k)) dx$$

$$s.t. \sum_{k=1}^{K} C^{mi}(x, Y = k) = 1. \tag{E5}$$

By maximizing the value function pointwisely and applying Lagrange multipliers, we obtain the following problem:

$$\{C^{mi*}(x, Y = 1), \ldots, C^{mi*}(x, Y = K)\}$$

$$= \arg \max_{C^{mi}(x, Y=1), \ldots, C^{mi}(x, Y=K)} \sum_{k=1}^{K} Q(x|Y = k) \log(C^{mi}(x, Y = k))$$

$$+ \lambda(\sum_{k=1}^{K} C^{mi}(x, Y = k) - 1). \tag{E6}$$

Setting the derivative of (E6) w.r.t. $C^{mi}(x, Y = k)$ to zeros, we obtain

$$C^{mi*}(x, Y = k) = -\frac{Q(x|Y = k)}{\lambda}. \tag{E7}$$

We can solve for the Lagrange multiplier $\lambda$ by substituting (E7) into the constraint $\sum_{k=1}^{K} C^{mi}(x, Y = k) = 1$ to give $\lambda = -\sum_{k=1}^{K} Q(x|Y = k)$. Thus we obtain the optimal solution

$$C^{mi*}(x, Y = k) = \frac{Q(x|Y = k)}{\sum_{k'=1}^{C} Q(x|Y = k')}. \tag{E8}$$

$\square$

Now we are ready the prove the theorem. If we add $K \log K$ to $U(G)$, we can obtain:

$$U(G) + K \log K$$

$$= \sum_{k=1}^{K} \mathbb{E}_{X \sim Q(X|Y=k)} \left[ \log \frac{Q(X|Y = k)}{\sum_{k'=1}^{K} Q(X|Y = k')} \right] + K \log K$$

$$= \sum_{k=1}^{K} \mathbb{E}_{X \sim Q(X|Y=k)} \left[ \log \frac{Q(X|Y = k)}{\frac{1}{K} \sum_{k'=1}^{K} Q(X|Y = k')} \right]$$

$$= \sum_{m=1}^{K} KL\left(Q(X|Y = k) \Big\| \frac{1}{K} \sum_{k=1}^{K} Q(X|Y = k')\right). \tag{E9}$$

By using the definition of JSD, we have

$$U(G) = -K \log K + K \cdot \text{JSD}(Q_{X|Y=1}, \ldots, Q_{X|Y=K}). \tag{E10}$$

Since the Jensen-Shannon divergence among multiple distributions is always non-negative, and zero if they are equal, we have shown that $U^* = -K \log K$ is the global minimum of $U(G)$ and that the only solution is $Q_{X|Y=1} = Q_{X|Y=2} = \cdots = Q_{X|Y=K}$. $\square$

## S3. Proof of Theorem 3

According to the triangle inequality of total variation (TV) distance, we have

$$d_{TV}(P_{XY}, Q_{XY}) \leq d_{TV}(P_{XY}, P_{Y|X}Q_X) + d_{TV}(P_{Y|X}Q_X, Q_{XY}). \tag{E11}$$

Using the definition of TV distance, we have

$$
\begin{aligned}
d_{TV}(P_{Y|X}P_X, P_{Y|X}Q_X) &= \frac{1}{2}\int |P_{Y|X}(y|x)P_X(x) - P_{Y|X}(y|x)Q_X(x)|\mu(x,y) \\
&\overset{(a)}{\leq} \frac{1}{2}\int |P_{Y|X}(y|x)|\mu(x,y)\int |P_X(x) - Q_X(x)|\mu(x) \\
&\leq c_1 d_{TV}(P_X, Q_X),
\end{aligned} \tag{E12}
$$

where $P$ and $Q$ are densities, $\mu$ is a ($\sigma$-finite) measure, $c_1$ is an upper bound of $\frac{1}{2}\int |P_{Y|X}(y|x)|\mu(x,y)$, and (a) follows from the Hölder inequality.

Similarly, we have

$$
d_{TV}(P_{Y|X}Q_X, Q_{Y|X}Q_X) \leq c_2 d_{TV}(P_{Y|X}, Q_{Y|X}), \tag{E13}
$$

where $c_2$ is an upper bound of $\frac{1}{2}\int |Q_X(x)|\mu(x)$ . Combining (E11), (E12), and (E13), we have

$$
\begin{aligned}
d_{TV}(P_{XY}, Q_{XY}) &\leq c_1 d_{TV}(P_X, Q_X) + c_2 d_{TV}(P_{Y|X}, Q_{Y|X}) \\
&\leq c_1 d_{TV}(P_X, Q_X) + c_2 d_{TV}(P_{Y|X}, Q_{Y|X}^c) + c_2 d_{TV}(Q_{Y|X}, Q_{Y|X}^c).
\end{aligned} \tag{E14}
$$

According to he Pinsker inequality $d_{TV}(P,Q) \leq \sqrt{\frac{KL(P||Q)}{2}}$ [1], and the relation between TV and JSD, *i.e.*, $\frac{1}{2}d_{TV}(P,Q)^2 \leq JSD(P,Q) \leq 2d_{TV}(P,Q)$ [2], we can rewrite (E14) as

$$
JSD(P_{XY}, Q_{XY}) \leq 2c_1\sqrt{2JSD(P_X, Q_X)} + c_2\sqrt{2KL(P_{Y|X}||Q_{Y|X}^c)} + c_2\sqrt{2KL(Q_{Y|X}||Q_{Y|X}^c)}. \tag{E15}
$$

## S4. 1D MoG synthetic Data

## S4.1. Experimental Setup

For all the networks in AC-GAN, Projection cGAN, and our TAC-GAN, we adopt the three layer Multi-Layer Perceptron (MLP) with hidden dimension 10 and Relu [3] activation function. The only difference is the number of input and output nodes. We choose Adam [4] as the optimizer and set the learning rate as 2e-4 and the hyperparameter of Adam as $\beta = (0.0, 0.999)$. We train 10 steps for $D$, $C$, and $C^{mi}$ and 1 step for $G$ in every iteration. The batch size is set to 256.

 **S4.2. More Results**

Figure 1: Change distance between the means of adjacent 1-D Gaussian Components, in this figure, all models adopt cross entropy loss.

Figure 2: Change distance between the means of adjacent 1-D Gaussian Components, in this figure, all models adopt hinge loss.

# S5. 2D MoG Synthetic Data

Figure 3: Change distance between the means of adjacent 2-D Gaussian Components in x-axis, in this figure, all models adopt cross entropy loss.

## S6. Overlapping MNIST

### S6.1. Experimental Setup

For the network settings, the $G$ network consists of three layers of Res-Block and relies on Conditional Batch Normalization (CBN) [5] to plug in label information. The network structure of $D$ mirrors $G$ network without CBN. To stabilize training, $D, C, C^{mi}$ share the the convolutional layers and differ in the fully-connected layers. The chosen dimension of latent $z$ is 128 and optimizer is Adam with learning rate $lr$=2e-4 and $\beta = \{0.0, 0.999\}$ for both $G$ and $D$ networks. Each iteration contains 2 steps of $D, C, C^{mi}$ training and 1 step of $G$ training. The batch size is set to 100.

### S6.2. More Results

In this experiment, we fix the training data and change the weight of classifier from $\lambda_c = 0.5$ to $\lambda_c = 3.0$ with step 0.5 for our model TAC-GAN and AC-GAN. For AC-GAN, when the value of $\lambda_c$ becomes larger, the proportion of the generated digit '0', which is the overlapping digit, goes smaller. However, our model is still able to replicate the true distribution.

Figure 4: More generated results for the overlapping MNIST dataset.

## S7. CIFAR100

### S7.1. Experimental Setup

Due to the complexity and diversity of this dataset, we apply the latest SN-GAN [6] as our base model, the implementation is based on Pytorch implementation of Big-GAN [7] and SN layer is added to both $G$ and $D$ networks [8]. On this dataset, there is no need to add Self-Attention layer [8] and only three Res-Blocks layers are applied due to the low resolution as $32 \times 32$. As done by SN-GAN [6], we replace the loss term ⓐ by the hinge loss in order to stabilize the GAN training part. For all evaluated methods, the batch size is 100 and total number of training iterations is 60K. The optimizer parameters are identical to those used in the overlapping MNIST experiment.

### S7.2. PAC-GAN improvement

pacGAN is a great method that significantly increases11the performance of AC-GAN, though the performance is still lower than our method in terms of both scores and visual12quality. This indicates that the drawbacks in AC-GAN loss cannot be fully addressed by pacGAN. We can see that13combining pacGAN and TAC-GAN increases the performance, suggesting that pacGAN and TAC-GAN are compatible.

Figure 5: Generated Images

| MetricMethod | Ours | pacGAN4+Ours | AC-GAN | pacGAN4+AC-GAN |
|---|---|---|---|---|
| IS | $9.34 \pm 0.077$ | $\mathbf{9.85 \pm 0.116}$ | $5.37 \pm 0.064$ | $8.54 \pm 0.143$ |
| FID | 7.22 | **6.79** | 82.45 | 20.94 |

tableIS and FID scores

### S7.3. More Results

We show the generated samples for all classes in Figure 6 and report the FID and LPIPS scores for each class in Figure 7 and Figure 8, respectively.

AC-GAN, $\lambda_c = 0.2$

TAC-GAN , $\lambda_c = 1.0$

Projection cGAN

Figure 6: 100 classes of CIFAR100 generated samples, we choose the classifier weight $\lambda_c = 0.2$ for AC-GAN model.

Figure 7: The FID score is reported for each class on CIFAR100 generated data, lower is better. The y axis denotes class label and x axis denotes FID score.

Figure 8: The LPIPS score is reported for each class on CIFAR100 generated data, larger values means better variance inner class. The y axis denotes class label and x axis denotes LPIPS score.

# S8. ImageNet1000

## S8.1. Experimental Setup

We adopt the full version of Big-GAN model architecture as the base network for AC-GAN, Projection cGAN, and TAC-GAN. In this experiment, we apply the shared class embedding for each CBN layer in $G$ model, and feed noise $z$ to multiple layers of $G$ by concatenating with class embedding vector. We use orthogonal initialization for network parameters [7]. In addition, following [7], we add Self-Attention layer with the resolution of 64 for ImageNet. Due to limited computational resources, we fix the batch size to 256. To boost the training speed, we only train one step for $D$ network and one step for $G$ network.

## S8.2. More Results

Figure 9: In this figure, we randomly select some generated samples from 1000 classes. It contains birds, snakes, bug, dog, food, scene, etc. Our model shows a very competitive fidelity and diversity. Generative models are all trained on ImageNet1000 and the image resolution is $128 \times 128$.

Figure 10: The FID score is reported for each class on ImageNet1000 generated data, we randomly select 100 classes from our generated samples for comparison between our model TAC-GAN and Projection cGAN. The method with a lower FID score is better. The y axis denotes class label and x axis denotes FID score.

Figure 11: The LPIPS score is reported for each class on ImageNet1000 generated data. we randomly select 100 classes from our generated samples for comparison between our model TAC-GAN and Projection cGAN, higher LPIPS socre means larger intra-class variance. The y axis denotes class label and x axis denotes LPIPS score.

## S9. VGGFace200

### S9.1. Experimental Setup

We adopt the full version of Big-GAN model architecture as the base network for AC-GAN, Projection cGAN, and TAC-GAN. In this experiment, we apply the shared class embedding for each CBN layer in $G$ model, and feed noise $z$ to multiple layers of $G$ by concatenating with class embedding vector. We use orthogonal initialization for network parameters [7]. In addition, following [7], we add Self-Attention layer with the resolution of 32 for VGGFace. Due to limited computational resources, we fix the batch size to 256. In this setting, we train two steps for $D$ network and two steps for $G$ network. The only difference of the networks applied on ImageNet and VGGFace is that the network on ImageNet has one additional up-sampling block and one more down-sampling block added to $G$ and $D$ Networks to accommodate higher resolution.

### S9.2. More Results

Figure 12: In this figure, we randomly select some generated samples for illustration. All the generative models are trained on 200 classes on the randomly sampled 200 classes from the VGGFace2 dataset.

Figure 13: The LPIPS score is reported for each class on VGGFace200 generated data. we randomly select 100 classes from our generated samples for comparison between our model TAC-GAN and Projection cGAN, higher LPIPS score means larger intra-class variance. The y axis denotes class label and x axis denotes LPIPS score.