[Reviews · NeurIPS 2019]

Reviewer 1



### After going through the author's rebuttal, I am improving my score. ##### Original: The work brings in a good amount of originality in understanding AC-GAN and provides an improvement motivated by theory. Though the analysis uses well-known tools, it helps to understand AC-GAN. Quality: The paper is well written and results are explained clearly. Significance: Since the given method improves upon Projection GAN in only some cases, and the fact that Projection GAN is simpler than the proposed method, its significance is limited.

Reviewer 2



I have read the authors' rebuttal. I am satisfied with the answers. I will keep my rating at 8. ---------- Questions / criticisms / suggestions: - I see that in your work you present ACGAN as being a particular instantiation of a cGAN (i.e. line 33), but I see these are two separate algorithms that do different things (e.g. see Figure 1 in the pcGAN paper). For instance, in cGAN the discriminator d(x,y) is estimating p(x|y)p(y) (which the generator tries to match with its conditional q(x|y) ), whereas in ACGAN it is implicitly estimating p(y|x)p(x), where p(y|x) is the auxiliary classifier and p(x) is the discriminator. It would be important to make this clear since these techniques are sufficiently different from each other. (An example of this is in the multi-label prediction case: suppose y_i is a binary variable {0,1} and i in {1,...k} for k classes (an example of this is the CelebA dataset). In cGAN, the discriminator learns p(x|y) = p(x|y1,...yk) (i.e. x is conditioned on joint y), whereas in ACGAN the discriminator learns p(y|x) = p(y1|x)...p(yk|x) (all y_i's are conditionally independent, given x). - In my own experience, I have noticed low intra-class diversity on cGANs (i.e., not ACGANs), so it seems to me like a similar theoretical analysis could be done on this as well. What are your thoughts on this, and have you also noticed low intra-class diversity for cGAN? - Regarding footnote at the end of page 4, what happens if you do not use biased batch sampling? Does the method still work well in practice? - How did you compute the FID in this case? Do you compute the so called 'intra-FID' which is used in the pcGAN paper? (E.g. compute FID for each class, and average these. See https://github.com/pfnet-research/sngan_projection/issues/34) Or did you simply compute one big FID over all the real and generated samples? You should describe how you computed this. In the case of the former, you could compute standard deviations for those FID numbers and add them to Table 1. I think this would make it much clearer to what extent your proposed method addresses the intra-class mode dropping problem. - For Table 1, did you run any repeat experiments? The standard deviations in the Inception scores usually come from one model (since the actual algorithm computes a mean over subsets of samples generated, hence why there is an uncertainty estimate), but you should average these over repeat experiments. In my experience training these models, there can be quite a bit of variance in IS *between* repeat experiments. If you haven't done this, I highly recommend you do so (within reason, of course, I realise these models can take a long time to train). - Spelling error, line 31, change 'lean' -> 'learn', and line 144 'combing' -> 'combining' - While this is undoubtedly beyond the scope of your work, it seems like one way to avoid this issue entirely is to train a bidirectional model like a VAE-GAN, since the model is bijective. In this case you don't have to worry about there being low intra-class diversity, and furthermore, because of the GAN aspect, the samples won't look blurry (like in regular autoencoders). This I feel dilutes the significance of the issue that this paper addresses. I would like to know your thoughts on this. In summary: - Originality: this work is original to the best of my knowledge. - Quality: the paper is well written, and there is sufficient empirical evaluation, on both real and toy datasets, though the authors could do a few things to make the results even more convincing. Multiple quantitative metrics were used as well, which is nice. - Clarity: the paper is easy to read, though some clarification on the difference between cGAN and ACGAN would be good, as well as how certain metrics were computed in the results section (IS and FID). - Significance: the paper addresses a well-known issue in class-conditional generation in GANs and proposes a solution. It seems likely that others would be open to adopting this idea / implementing it since it's an easy implementation detail: simply extend the minimax game by adding a 2nd classifier which only predicts the labels of fake samples.

Reviewer 3



As most of the concerns have been addressed by the authors' feedback, I upgrade my score. ############################################################# Auxiliary classifier GAN, as a popular conditional GAN, tends to generate near-identical images for most classes as the number of labels increases. In this paper, the authors have an in-depth discussion on the source of the low-diversity problem. In particular, the authors suggest that the AC-GAN misses an important negative conditional entropy term, so that it cannot faithfully minimize the divergence between real and generated conditional distribution. Based on this insightful observation, a new twin auxiliary classifiers GAN has been developed in this paper. The authors have clearly analyzed the problem and developed a novel solution with solid theoretical supports. The proposed algorithm has been evaluated on both synthetic data and real-world data, and demonstrated impressive performance in terms of most metrics.

[Author Response · NeurIPS 2019]

We would like to thank all reviewers evaluating the paper, and will fully address all the review concerns in the revision.

**Re R#1: The given method improves upon Projection cGAN (PcGAN) in only some cases**. From the current IS
and FID results, we are only inferior to PcGAN on the ImageNet dataset. Especially, our method is much better than
PcGAN on the VGG face dataset. We have recently tested VGG face using 2000 classes. The IS score of TAC-GAN
and PcGAN are $109.04\pm2.44$ and $79.51\pm1.03$. The FID score of TAC-GAN and PcGAN are 13.79 and 22.42. The
results suggest that our method is advantageous on fine-grained datasets in which classes are more close to each other.

**Projection cGAN is simpler than the proposed method**. Since the four players share the convolutional layers, our
method only adds a FC layer to AC-GAN. Compared to projection cGAN, our method only has a slight increase in
computational load because of calculation of two additional losses.

**pacGAN + ACGAN**. Thanks for the nice suggestion. As suggested by R#1, we combine pacGAN with AC-GAN and
our TAC-GAN, and the results are reported in Table 1 and Fig 1. pacGAN is a great method that significantly increases
the performance of AC-GAN, though the performance is still lower than our method in terms of both scores and visual
quality. This indicates that the drawbacks in AC-GAN loss cannot be fully addressed by pacGAN. We can see that
combining pacGAN and TAC-GAN increases the performance, suggesting that pacGAN and TAC-GAN are compatible.

| Metric \ Method | Ours | pacGAN4+Ours | AC-GAN | pacGAN4+AC-GAN |
|---|---|---|---|---|
| IS | $9.34 \pm 0.077$ | $\mathbf{9.85 \pm 0.116}$ | $5.37 \pm 0.064$ | $8.54 \pm 0.143$ |
| FID | 7.22 | **6.79** | 82.45 | 20.94 |

Figure 1: Generated Images          Table 1: IS and FID scores

**Re R#2:** We would like to thank R#2 for very detailed comments. **About Difference between ACGAN and cGAN.**
We consider AC-GAN as a particular type of cGAN, as suggested also in the PcGAN paper. This is because the generator
of AC-GAN and usual cGANs models the conditional distribution $p(x|y)$. The difference is how the discriminators
match joint distributions of $(X, Y)$ between generated and real data. This can be done in several ways as suggested
in Figure 1 in the PcGAN paper (The tile of this figure is " Discriminator models for conditional GANs"). The usual
cGAN concatenates $X$ (or features of X) with $Y$ and then use standard GAN loss to match joint distributions $p(x, y)$
(real) and $q(x, y)$ (fake). AC-GAN and PcGAN make use of the factorization $p(x, y) = p(y|x)p(x)$ to match the two
factors separately, but AC-GAN has a imperfect loss that fails to match $p(y|x)$ with $q(y|x)$.

**Low intra-class diversity for cGAN.** The intra-class diversity of usual cGANs (concatenation) cannot be explained by
the theorems in our paper. Our Theorems addresses the problems in AC-GAN loss. The usual cGANs have theoretically
correct losses, and there are no clear answers to their bad performance. One hypothesis is that directly match the joint
distributions of $(x, y)$ by concatenation is hard. Both PcGAN and our method suggest that using the factorization
$p(x, y) = p(y|x)p(x)$ and taking advantages of special structures in $p(y|x)$ is more effective.

**What if not using biased batch sampling.** If not using biased sampling, mutual information and JSD are not
computationally equivalent, which will degrade the performance.

**How is FID computed.** We used the scripts in the BigGAN repository to calculate the scores. The FID scores were
calculated on the entire dataset. We have also provided FID scores of each class in the Supplement.

**Repeat experiments in Table 1** Following the procedures in PcGAN and the state-of-the-art BigGAN method, we did
not repeat the experiments. We agree with R#2 that repeating the experiments is definitely much better to compare
different methods, but we have limited HPC resources to repeat the experiments during a short rebuttal period.

**VAE-GAN** The generative & inference networks in bidirectional GANs (eg, VAE-GAN) construct the cycle-consistency
term, which provides a bound to increase the entropy of the generated samples, thus improving (intra-class) diversity,
as shown in Lemma 1 and Figure 3 of [1]. Cycle-consistency is often considered in unsupervised learning, while we
propose the auxiliary classifier to explicitly improve intra-class diversity in class-conditional generation.

**Re R#3: why choose KL over JSD.** We choose KL because its nice connection to the cross-entropy loss. Using KL
will make the algorithm much simpler because we only need classification losses to match the conditional distributions.

**More clear explanation of the missing conditional entropy term.** Yes, when updating the classifier $C$, the conditional
entropy can be considered as a constant term. But when updating the generator $G$, it cannot be considered as constant
because $G$ is involved in this term. We will discuss these two situations in more detail in the updated version.

**Regarding complexity and stability** Please refer to the second answer to R#1 for explanationo of complexity. In all
our experiments, we did not specifically tune the hyperparameter $\lambda_c$ and we found our method pretty stable, as shown
in Fig 6 in the main paper.

[1] Chunyuan Li, Ke Bai, Jianqiao Li, Guoyin Wang, Changyou Chen, and Lawrence Carin. Adversarial learning of
sampler based on an unnormalized distribution.arXiv preprint arXiv:1901.00612, 2019.

[Meta-Review · NeurIPS 2019]

This paper addresses a well-known issue in class-conditional generation in GANs and proposes a solution. The reviewers are satisfied with how the authors addresses their concerns in the rebuttal, and agree on accepting the paper. The reviewers agree that the paper addresses an interesting challenge present in a popular GAN architecture with auxiliary classifiers. A solution to this problem could potentially be widely used in GANs with auxiliary classifiers such as InfoGAN, ACGAN, etc.